# Lenvatinib-Loaded Poly(lactic-co-glycolic acid) Nanoparticles with Epidermal Growth Factor Receptor Antibody Conjugation as a Preclinical Approach to Therapeutically Improve Thyroid Cancer with Aggressive Behavior

**DOI:** 10.3390/biom13111647

**Published:** 2023-11-13

**Authors:** Giovanna Revilla, Nuseibah Al Qtaish, Pablo Caruana, Myriam Sainz-Ramos, Tania Lopez-Mendez, Francisco Rodriguez, Verónica Paez-Espinosa, Changda Li, Núria Fucui Vallverdú, Maria Edwards, Antonio Moral, José Ignacio Pérez, Juan Carlos Escolà-Gil, José Luis Pedraz, Idoia Gallego, Rosa Corcoy, María Virtudes Céspedes, Gustavo Puras, Eugènia Mato

**Affiliations:** 1Research Biomedical Institute (IIB) Sant Pau, C/Sant Quintí 77, 08041 Barcelona, Spain; grevilla@santpau.cat (G.R.); pcaruana@santpau.cat (P.C.); pacorl97@outlook.es (F.R.); clix@santpau.cat (C.L.); nuriafvg@gmail.com (N.F.V.); medwards@santpau.cat (M.E.); jescola@santpau.cat (J.C.E.-G.); rcorcoy@santpau.cat (R.C.); 2Departament of Biochemistry and Molecular Biology, Universitat Autònoma de Barcelona, C/Antoni M. Claret 167, 08025 Barcelona, Spain; 3Department of Endocrinology and Nutrition, Hospital de la Santa Creu i Sant Pau, 08041 Barcelona, Spain; 4Networking Research Centre of Bioengineering, Biomaterials and Nanomedicine (CIBER-BBN), C/Monforte de Lemos 3-5, 28029 Madrid, Spain; nusaiba.qtaish@gmail.com (N.A.Q.); myri.2694@gmail.com (M.S.-R.); tania.lopez@ehu.eus (T.L.-M.); amoral@santpau.cat (A.M.); joseluis.pedraz@ehu.eus (J.L.P.); idoia.gallego@ehu.eus (I.G.); gustavo.puras@ehu.eus (G.P.); 5NanoBioCel Research Group, Laboratory of Pharmacy and Pharmaceutical Technology, Faculty of Pharmacy, University of the Basque Country (UPV/EHU), Paseo de la Universidad 7, 01006 Vitoria-Gasteiz, Spain; 6Pharmacy Department, College of Pharmacy, Amman Arab University, P.O. Box 2234, Amman 11953, Jordan; 7Bioaraba, NanoBioCel Research Group, 01009 Vitoria-Gasteiz, Spain; 8Department Clinical Biochemistry, School of Medicine, Pontificia Universidad Católica del Ecuador (PUCE), Av. 12 de Octubre 1076 y Roca, Quito 17012184, Pichincha, Ecuador; epaez387@puce.edu.ec; 9Department of General Surgery, Hospital de la Santa Creu i Sant Pau, C/Sant Quintí 89, 08041 Barcelona, Spain; jperezg@santpau.cat; 10CIBER de Diabetes y Enfermedades Metabólicas Asociadas (CIBERDEM), C/Monforte de Lemos 3-5, 28029 Madrid, Spain

**Keywords:** thyroid, nanoparticles, EGFR, lenvatinib

## Abstract

Background: Lenvatinib, a tyrosine kinase inhibitor (TKI) approved for the treatment of progressive and radioactive iodine (RAI)-refractory differentiated thyroid cancer (DTC), is associated with significant adverse effects that can be partially mitigated through the development of novel drug formulations. The utilization of nanoparticles presents a viable option, as it allows for targeted drug delivery, reducing certain side effects and enhancing the overall quality of life for patients. This study aimed to produce and assess, both in vitro and in vivo, the cytotoxicity, biodistribution, and therapeutic efficacy of lenvatinib-loaded PLGA nanoparticles (NPs), both with and without decoration using antibody conjugation (cetuximab), as a novel therapeutic approach for managing aggressive thyroid tumors. Methods: Poly(lactic-co-glycolic acid) nanoparticles (NPs), decorated with or without anti-EGFR, were employed as a lenvatinib delivery system. These NPs were characterized for size distribution, surface morphology, surface charge, and drug encapsulation efficiency. Cytotoxicity was evaluated through MTT assays using two cellular models, one representing normal thyroid cells (Nthy-ori 3-1) and the other representing anaplastic thyroid cells (CAL-62). Additionally, an in vivo xenograft mouse model was established to investigate biodistribution and therapeutic efficacy following intragastric administration. Results: The NPs demonstrated success in terms of particle size, polydispersity index (PDI), zeta potential, morphology, encapsulation efficiency, and cetuximab distribution across the surface. In vitro analysis revealed cytotoxicity in both cellular models with both formulations, but only the decorated NPs achieved an ID50 value in CAL-62 cells. Biodistribution analysis following intragastric administration in xenografted thyroid mice demonstrated good stability in terms of intestinal barrier function and tumor accumulation. Both formulations were generally well tolerated without inducing pathological effects in the examined organs. Importantly, both formulations increased tumor necrosis; however, decorated NPs exhibited enhanced parameters related to apoptotic/karyolytic forms, mitotic index, and vascularization compared with NPs without decoration. Conclusions: These proof-of-concept findings suggest a promising strategy for administering TKIs in a more targeted and effective manner.

## 1. Introduction

Well-differentiated thyroid carcinomas (WDTCs) are generally considered malignant tumors with a positive prognosis. However, it is crucial to recognize that a small percentage, typically between 5% and 15%, may display aggressive behavior and become unresponsive to radioiodine (RAI) treatment, leading to the development of refractory tumors. Additionally, dedifferentiated thyroid cancer (TC) and anaplastic thyroid cancer (ATC) are associated with a poorer prognosis in comparison with well-differentiated thyroid cancer (WDTC). One of the key characteristics of dedifferentiated and aggressive TC that is refractory to RAI treatment is the loss of the ability to uptake ^131^I. This loss of iodine uptake represents a significant survival rate reduction in patients (60 to 70%) five years after diagnosis [1,2,3]. In these tumors subtypes, including ATC, poorly differentiated thyroid carcinoma (PDTC), and RAI-refractory WDTC patients, the identification of the oncogenic driver mutations, able to activate specific kinases, has allowed the use of tyrosine kinase inhibitors (TKIs) as a therapeutic strategy [4,5,6,7]. For instance, the TKI lenvatinib demonstrated a significantly longer median progression-free survival (PFS) compared with both sorafenib and the placebo group; PFS time for lenvatinib was 18.3 months, whereas it was only 3.6 months for the placebo group (*p* < 0.001). In comparison, sorafenib had a PFS of 10.8 months, and the placebo group had one of 5.8 months (both with *p* < 0.0001). This outstanding performance has positioned lenvatinib as the preferred choice among oral antiangiogenic multitargeted TKIs [8,9,10,11,12].

This multitargeted TKI works against vascular endothelial growth factor receptors (VEGFRs) 1, 2, and 3; Fibroblast growth factor receptors (FGFRs) 1 through 4; Platelet-derived growth factor receptor alpha (PDGFRA); Ret Proto-Oncogene (RET); and the KIT Proto-Oncogene, Receptor Tyrosine Kinase signaling network, which are all involved in tumoral angiogenesis processes [13,14,15,16]. Nevertheless, in most treated patients, a dose reduction or treatment discontinuation may be necessary due to a high frequency of adverse events associated with VEGF-targeted therapies, including hypertension, diarrhea, fatigue or asthenia, decreased appetite, and weight loss [12,13,17]. Moreover, the Epidermal Growth Factor Receptor (EGFR) represents a promising and valid therapeutic target in solid tumors, including ATC, that exhibits overexpression of this receptor [18]. Cetuximab is a human–murine chimeric EGFR-targeted monoclonal antibody with increased specificity for the extracellular domain of human EGFR. Its mechanism of action involves inhibiting EGFR signaling in cells, thereby disrupting the normal function of the receptor [19,20]. The use of nanosystems to deliver drugs could be considered a good strategy for reducing the adverse effects of these drugs. The poly(lactic-co-glycolic acid) (PLGA) nanosystem is considered one of the most favorable choices, owing to its biodegradability and efficient clearance via the pulmonary and renal routes, as previously documented [21,22,23,24]. Through our previous investigations with sorafenib-loaded PLGA-nanoparticles (NPs), we observed that NPs decorated with cetuximab exhibited a more pronounced and sustained cytotoxic effect over time in the anaplastic thyroid cell line CAL-62, compared with the normal thyroid cell line Nthy-ori 3-1. These findings suggest a promising targeted strategy to treat WDTC with poor prognosis [25]. Considering the advantages of lenvatinib to sorafenib in the efficacy, safety, and survival of patients, the objective of this study was to optimize the formulation of lenvatinib-loaded PLGA nanoparticles (lenva-NPs), both with and without cetuximab decoration. Furthermore, we aimed to assess their in vitro cytotoxicity and, as a proof of concept, to evaluate their biodistribution and therapeutic effectiveness in a xenograft model of anaplastic thyroid carcinoma.

## 2. Materials and Methods

### 2.1. Materials

The lenvatinib (E7080) (10 mg) was supplied by Selleckchem (Deltaclon, Madrid, Spain), and the cetuximab (anti-EGFR; 5 mg/mL) was obtained from Selleckchem, Deltaclon (Madrid, Spain). The polymer poly (D, L-lactide-co-glycolide; PLGA, Resomer^®^ RG 503, Mw 33.900), with a copolymer ratio of 50:50 (lactic/glycolic) and an intrinsic viscosity of 0.8 dL/g in CHCl3, was supplied by Boehringer Ingelheim K.G. (Ingelheim, Germany). Polyvinyl alcohol (PVA, MW 30,000–70,000), dichloromethane, dimethyl sulfoxide, 1-ethyl-3-(3-dimethylaminopropyl) carbodiimide hydrochloride (EDC), and N-hydroxysulfosuccinimide (sulfo-NHS) were supplied by Sigma Chemical Co. (St. Louis, MO, USA). 1,1′-Dioctadecyl-3,3,3′,3′-Tetramethylindotricarbocyanine Iodide (DiIC) was supplied by Thermo Fisher Scientific (Waltham, MA, USA). A micro BCA assay kit was purchased from Pierce by Teknovas (Bilbao, Spain). All other chemicals were of analytical grade and were supplied by Panreac S.A. (Barcelona, Spain).

### 2.2. Preparation of Lenvatinib-Loaded PLGA Nanoparticles (Lenva-NPs)

The oil-in-water (O/W) single emulsion solvent evaporation method was used in the preparation of the lenva-NPs, with a slight modification. To guarantee perfect solubility, 200 mg of lenvatinib was dissolved in 200 μL of dimethyl sulfoxide and rapidly sonicated for 15 s. This solution was added to 3.800 μL of dichloromethane containing 200 mg of PLGA to obtain the final oil solution (5% PLGA, *w*/*v*), which was then poured into 20 mL of an 8% polyvinyl alcohol aqueous solution and sonicated for 100 s at 50 W (Branson Sonifier 250, Branson Ultrasonics™, Shanghai, China) over an ice bath. To facilitate the evaporation of volatile organic solvents from the NPs to the external phase, a 2% isopropanol solution was added, and the system was agitated for 2 h at room temperature on a magnetic stir plate. Ultra-centrifugation at 30,000 rpm, 4 °C for 20 min (Sorvall Legend X1R Centrifuge, Thermo Fisher Scientific, Osterode am Harz, Germany) was employed to separate the NPs, which were then washed with double distilled water (the process was repeated three times). The recovered NP suspension was freeze-dried with 5% trehalose (*w*/*v*) for 24 h (Lyo Beta 15, Telstar, Tarrasa, Spain).

### 2.3. Characterization of Lenva-NPs: Particle Size, Polydispersity Index (PDI), Zeta Potential Measurements, and Morphology Studies (TEM)

The lenva-NPs were analyzed using a Malvern Zetaizer Nano ZS in terms of particle size, dispersity index, and zeta potential (Malvern Instrument, Worcestershire, UK). Dynamic light scattering was used to determine the particle size based on backscatter detection optics at 173°. In brief, 1 mg of lenva-NPs was resuspended in 1 mL of a 0.1 mM NaCl aqueous solution. The refractive index (1.33) and viscosity (0.89) of ultrapure water at 25 °C were utilized for manual data analysis. All measures were performed in triplicate, and each triplicate was measured 11 times to obtain the average. Cumulative analysis was used to determine the particle size reported as the hydrodynamic diameter. The zeta potential was measured using laser Doppler velocimetry. For the zeta analysis, the lenva-NPs were resuspended using disposable folded capillary cells, as in the size measurement procedure. Similarly, measurements of the zeta potential were performed in triplicate using the manual mode, with 20 measurements in each run. The Hückel approximation was utilized to support the zeta potential calculation. Only data that met the software program’s v. 7.11 (DTS 5.0, Zetasizer Nano System) quality criteria were included in the study.

### 2.4. Transmission Electron Microscope (TEM) Studies

TEM examination was used to observe the morphology of the lenva-NPs. Briefly, the samples were adhered to glow-discharged carbon-coated grids for 60 s. To boost the contrast, the remaining liquid was collected by blotting on a paper filter and dyed with 2% uranyl acetate for 60 s at a pH of 6.0. Samples were observed using a microscope, TECNAI G2 20 TWIN (FEI, Eindhoven, The Netherlands), operated in bright-field image mode at 200 KeV accelerating voltage. Using an Olympus SIS Morada digital camera, digital images were captured.

### 2.5. Assessing the Incorporation of Lenvatinib into Nanoparticles with HPLC

Using reversed-phase HPLC (RP-HPLC), the encapsulation efficiency of lenvatinib into PLGA NPs was assessed. Lenvatinib was analyzed on a YMC C18 (4.6 × 150 mm, 5 m) column. We utilized a mobile phase composed of HPLC-grade water and methanol (30/70 *v*/*v*), with the flow rate set at 0.6 mL/min, to elute the lenvatinib. Under vacuum filtration, the mobile phase was filtered using 0.22-micron nylon filters and degassed for 5 min in an ultrasonic water bath. As a diluent, the mobile phase was utilized. The injection volume was 20 μL, and the detector was calibrated at 240 nm. Both the auto-sampler and the column were kept at room temperature. The chromatographic program runtime was 7.0 min. The lenvatinib sample solution was prepared, and 10 mg of lenvatinib were accurately weighed and then deposited into a volumetric flask containing 10 mL of clean, dry volumetric solution. Next, about 2 mL of diluent (mobile phase) was added, and the mixture was sonicated to dissolve it completely. Finally, the volume was brought up to the appropriate mark using a diluent. An additional 10 mL of the aforesaid stock solution was pipetted into a 100 mL volumetric flask and diluted with diluent to the mark. Then, 10 μL of the standard and sample solutions of the lenvatinib drug were injected into the chromatographic system in triplicate, and the peak areas for the lenvatinib drug were measured. Using a formula, the assay percentage was calculated by comparing the peak area of the standard and sample chromatograms. The percentage of encapsulation efficiency was calculated by dividing the results of the subtraction by the theoretical lenvatinib value, using the following equation:$$EE(\%)=\frac{theoretical\;lenvatinib-free\;lenvatinib}{theoretical\;lenvatinib}\times 100$$

Lenvatinib drug content was expressed as the mass of incorporated lenvatinib (μg) per each mg of lyophilized lenva-NPs.

### 2.6. In Vitro Lenvatinib Release Studies

To study lenvatinib release from NPs, 10 mg of Lenva-NPs were added to test tubes containing 1 mL of PBS 0.1 M and 0.01% (*w*/*v*) tween 80 at a pH of 7.4. The suspensions of NPs were incubated at 37 °C with constant orbital rotation. The samples were spun at 30,000 rpm, 4 °C, for 20 min at regular intervals of up to 38 days. The supernatants were collected, and the released lenvatinib was quantified using RP-HPLC. The same volume utilized for the determination of lenvatinib was replaced with a new buffer. The experiment was conducted with three distinct batches.

### 2.7. Preparation of Immunonanoparticles (Lenva-NPs-Cetuximab)

EDC/sulfo-NHS cross-linking chemistry was used to attach the cetuximab monoclonal antibodies to the surface of the lenva-NPs; briefly, 40 mg of lenva-NPs was resuspended in 4 mL of 0.1 M, 0.5 M NaCl, pH 5.5 MES buffer. Then, 1.6 mg of EDC (final concentration 2 Mm) was applied directly to the nanoparticle suspension, yielding a 10-fold molar excess of EDC to PLGA. The rapid addition of 4.4 mg of sulfo-NHS (final concentration 5 Mm) to the lenva-NP suspension, followed by 30 min of agitation on a magnetic plate, allowed the reaction to occur. Then, 5 μL of 2-mercaptoethanol (final concentration 20 Mm) was added to inactivate the EDC. The sediment was then ultracentrifuged for 20 min at 30,000 rpm and 4 °C to remove the unreacted EDC, sulfo-NHS, and 2-mercaptoethanol. The process was repeated three times. Each time the sediment was washed with 1 mL of PBS 0.1 M, 0.15 M NaCl. To dissolve the pellet obtained after the last centrifugation, 1 mL of anti-EGFR solution (1 mg/mL in PBS 0.1 M) was added, agitated for 2 h at room temperature on a magnetic stir plate, and incubated overnight at 4 °C. The following day, to remove any unconjugated cetuximab, the suspension was ultracentrifuged for 20 min at 30,000 rpm and 4 °C. The process was repeated three times. Each time the sediment was washed with 1 mL of PBS 0.1 M, 0.15 M NaCl. The supernatant was collected for the determination of unconjugated anti-EGFR, and the resulting pellet was lyophilized using an aqueous solution of trehalose (5%, *w*/*v*) for 24 h.

### 2.8. Characterization of Lenvatinib-NP-Cetuximab Immunonanoparticles

The resulting immunonanoparticles were characterized in terms of particle size, PDI, and morphology, as previously described.

### 2.9. Quantification of Cetuximab Conjugated onto the Surface of Nanoparticles

Spectrophotometry was used to quantify the amount of unbound antibodies in the supernatant using a colorimetric microBCA protein assay kit (Infinite M200 microplate reader, Tecan Austria GmbH, Grödig, Austria). Lenva-NP-cetuximab (cetux) immunonanoparticles were obtained by subtracting free cetuximab from the total amount of cetuximab loaded (theoretical anti-EGFR). Therefore, the cetuximab content was represented as the mass of cetuximab incorporated (g) per mg of lyophilized nanoparticles. A sample taken from the supernatant of the lenva-NPs without cetuximab was used as a control. A standard curve of the cetuximab solution in a concentration range of 5 μg/mL to 100 μg/mL was used to compare the results.

### 2.10. Preparation of DilC 18 Fluorescently Labeled Nanoparticles

To determine the cellular uptake of the NPs, the fluorescent dye DilC18 was added to the dimethyl sulfoxide (DMSO) solution (0.05% *w*/*v*) instead of lenvatinib. A fluorescent PLGA NP produced previously (described in Section 2.2) was then added to obtain the DilC fluorescent NPs. The resulting DilC fluorescent-loaded and anti-EGFR conjugated DilC NPs were physiochemically characterized in terms of size, PDI, and morphology (Section 2.3).

### 2.11. Cell Lines

The human thyroid follicular epithelial cell line (Nthy-ori 3-1) was provided by Dr. Pilar Santisteban (CSIC, Madrid, Spain), and the human epithelial ATC cell line (CAL-62) was provided by Leibniz-Institute DSMZ GmbH (ACC 448). The short tandem repeats (STRs) of the Nthy-ori 3-1 cell line were analyzed following the manufacturer’s instructions using the protocols for the GeneAmp^®^ PCR System 2400 Thermal Cycler (Perkin Elmer/Applied Biosystems, Waltham, MA, USA), the ABI PRISM^®^310 Genetic Analyzer (Perkin Elmer/Applied Biosystems, Waltham, MA, USA), and the software package Genemapper 4.1 (Applied Biosystems, Carlsbad, CA, USA). The STR of the Nthy-ori 3-1 cell line was matched with the results reported in cellosaurus database (Bairoch A., 2018). The Nthy-ori 3.1 cell line was cultured in RPMI 1640 (with L-glutamine), supplemented with 10% fetal bovine serum and 2% streptomycin/penicillin. The CAL-62 was cultured in Dulbecco’s modified Eagle’s medium (DMEM; Gibco, Invitrogen, Carlsbad, CA, USA), supplemented with 10% heat-inactivated fetal calf serum (FCS; Gibco), 20 mM L-GlutaMaxI (Gibco, Invitrogen Carlsbad, CA, USA), and 1% penicillin-streptomycin (Gibco, Invitrogen Carlsbad, CA, USA).

### 2.12. In Vitro Cytotoxicity Studies of Lenva-NPs and Lenva-NP-Cetux

The in vitro cytotoxicity of the lenva-NP-cetux and lenva-NP formulations was assessed with 3-(4,5-dimethylthuazol-2-yl)-2,5-diphenyltetrazolium bromide assay (MTT) (Sigma, M6494, MO, USA). The cell lines (Nthy-ori 3-1 and CAL-62) were placed in 96-well plates with 200 μL of media at a density of 1500 cells per well. After 24 h of culture, the media were removed, and fresh media with the NPs (lenva-NPs-cetux and lenva-NPs; 20 μM) were added and incubated for 24 h. At the end of the incubation period, the media was gently aspirated, and two washes with PBS 1× at 37 °C were carried out to remove any non-internalized NPs. Subsequently, 50 µL of PBS 1× at 37 °C was added along with 10 μL of MTT (12 mM), and the cells were incubated for an additional 4 h at 37 °C. Subsequently, the formazan crystals formed were dissolved by shaking with 50 µL of dimethyl sulfoxide (DMSO; SIGMA D8418) added to each well, mixing thoroughly with the pipette, and incubating at 37 °C for 10 min. The absorbance of each well was measured at 540 nm using an ELISA microplate reader (xMark BIORAD). The cytotoxicity rate was calculated as a percentage using the following formula: A treated cells/A control cells x100, where A is the optical density of the formazan released by the cells. In each experiment, untreated cells were used as controls.

### 2.13. Biodistribution of NP and NP-Cetux Nanoparticles in a Mouse Xenograft Model

In order to evaluate the biodistribution of the nanoparticles, CAL62 cell lines were subcutaneously injecting into the right flank of the athymic Swiss nu/nu mice at (20 × 10^6^ cells /100 µL, per flank) (mice donors). The tumoral growth was monitored by measuring their size twice a week until it reached a volume of 500 mm^3^. At this point, the mice were euthanized, and the tumors were excised and cut into small pieces measuring 2–3 mm^3^ in size. Subsequently, these tissue fragments were used for subcutaneous implantation into five-week-old female athymic Swiss nu/nu mice, weighing 18–20 g (recipient host), following the procedure described by Céspedes et al. [26].

After three weeks of tumor implantation, the subcutaneous 300 mm^3^ tumor-bearing mice were randomized into three groups: (1) control (N= 5), (2) NPs (n = 5), and (3) NPs-cetux (n = 5), and the NPs (200 µg/100 µL) or vehicle (PBS buffer, 100 µL) were administrated intragastrically (by gavage) three times per week for 4 weeks. Mice body weights and tumor volumes were recorded twice per week. Tumors were measured using a caliper and the volumes were estimated according to the formula V = (ab^2^)/2, with a: length or longest diameter, and b: width or shortest diameter. Measurement began on the first day of treatment (day 0) and finished on day 22 (when the first tumor in the control group reached 1 cm in diameter). At this time, all mice were euthanized, and the tumor and organs were removed for analysis.

The ex vivo biodistribution fluorescent image analysis was conducted by measuring the fluorescent lifetime signals (FLIs) using an IVIS^®^ Spectrum (Perkin Elmer, Waltham, MA, USA), which were correlated with the amount of dyed NPs accumulated in each organ and tumoral tissue. The signal obtained was digitalized and normalized by organ size and expressed as radiant efficiency ((p/s/cm2/sr)/μW/cm^2^). FLI values were calculated by subtracting the autofluorescence from the negative control and were finally expressed as a tissue/tumor ratio to compare the biodistribution profile between the different NPs.

Data were expressed as a percentage of radiant efficiency twice after NP administration (5 and 48 h) in each organ. The excitation-emission filters used to measure the FLI of each NP, and nanoconjugates were between 710–745 and 780–800, respectively.

### 2.14. Mouse Xenograft Model: In Vivo Intragastric Administration of Nanoparticles with Lenvatinib and Nanoparticles with Lenvatinib and Decorated with Cetuximab

Small pieces of tumors previously obtained from animal donors were implanted into mice as described in Section 2.13. The mice were randomized and divided into three groups (N = 7 mice per group). When the subcutaneous (SC) tumor reached approximately 100–120 mm^3^, and they received 200 µg intragastric (i.g.) bolus of NPs (lenva-NPs) and lenva-NPs-cetux in PBS buffer three times/week over 4 weeks; the control mice received an i.g. bolus of PBS vehicle, based on the results obtained in previous biodistribution assays. The mice were euthanized, and their tumors and organs were collected for analysis, as shown in the graphical abstract figure. The sample size was calculated according to our experience in similar experiments [26].

The care and use of all mice in this study were under procedures approved by the IIB Sant Pau Animal Ethics Committee regulations in accordance with the Federation of European Laboratory Animal Science Associations (FELASA; protocol number 11180).

### 2.15. Histopathological and Histomorphometric Assessment and Immunohistochemistry Analysis of Tumor Necrosis and Proliferation Rates

The tumor and non-tumor tissues were formalin-fixed, paraffin-embedded, and cut in sections of 5 μm thickness, followed by hematoxylin–eosin staining and mounted in Permount (Fisher Scientific). To quantify the total histological necrosis in the tumors, three areas were defined: (1) total area of the tumor, (2) acellular regions considered necrotic tissue (appearing pale pink), and (3) cellular tissue. All of these regions were measured using the freehand tool in ImageJ software v.1.8.0. The percentage of necrosis was calculated by dividing the necrotic area by the total area of the tumor and multiplying it by 100. Apoptosis/karyolisis and mitotic counts in the tumors were performed in random 10 high power fields (HPFs) at ×400 magnification.

All data were analyzed with ImageJ/Fiji software and expressed in percentages. The vascularization status/index/rate (vascular density and vascular length density) analysis was performed using the ImageJ/Fiji 1.46r program, which is capable of automatically calculating vascular density metrics, where vascular density = vessel area/total area * 100%, and vascular length density = skeletonized vessel area/total area * 100% [27,28]. Immunohistochemical analysis for VGFR2 (anti-VEGF receptor 2 antibody ab39638) and Ki-67 Antigen (Clone MIB-1, M7240, DAKO) were performed in a DAKO Autostainer Link48 following the manufacturer’s instructions. All the images were acquired using a microscopy Olympus Bx51 with a DP72 digital camera and processed with CellD Imaging 3.3 software (Olympus Corporation, Tokyo, Japan).

### 2.16. Western Blotting

Western blot analysis was used to evaluate the protein expression levels of AIF (sc-13116, Santa Cruz Biotechnology, Inc., Heidelberg, Germany) and Citocrom C (EPR1327, Abcam, Cambridge, UK) in orthotopic thyroid tumor tissue. The tumor tissue was dispersed mechanically with RIPA buffer (50 mM Tris-HCl, pH 7.5; 150 mM NaCl; 1% NP40; 0.5% sodium deoxycholate; 0.1% SDS; 1 mM EDTA) supplemented with protease inhibitor cocktail (Roche Diagnostics, Minato City, Tokyo, Japan), phenylmethylsulfonyl fluoride (PMSF, Sigma, St. Louis, MO, USA), and sodium orthovanadate (Sigma, St. Louis, MO, USA). The supernatant was collected after centrifugation at 12,000× *g* for 15 min at 4 °C and a BCA protein assay reagent kit (ThermoFisher Scientific, Waltham, MA, USA) was used to obtain the protein concentrations. Afterwards, the protein extracts were mixed with a 4× Laemmli loading buffer and heated at 94 °C for 4 min. Then, 20 μg of protein was size-separated on a 10% TGX Stain-Free precast gel (Bio-Rad, Hercules, CA, USA), transferred to a 0.2 μm PVDF membrane (Bio-Rad, Hercules, CA, USA) and blocked with 3% dried milk in Tris-buffered saline containing 0.05% of Tween-20 (TBST buffer) for 15 min. Finally, membranes were incubated with the primary antibody (both at 1/200 dilution) overnight at 4 °C. Thereafter, the membranes were washed three times for 10 min with TBST buffer and re-incubated with the IgG HRP-conjugated secondary antibody for 1 h. Finally, the membranes were washed three times for 10 min with TBST buffer and analyzed using Immun-Star Western Chemiluminescence Kit (Bio-Rad, Hercules, CA, USA). TGX Stain-free gels were activated for 1 min after SDS-electrophoresis. Images were captured using a ChemiDoc XRS Gel Documentation System (Bio-Rad, Hercules, CA, USA) and Image Lab software (version 6.0.1, Bio-Rad, Hercules, CA, USA). Data normalization analysis for each protein band was performed with the stain-free gel image saved, and the background was adjusted in such a way that the total background was subtracted from the sum of the density of all the bands in each lane.

### 2.17. Statistical Analysis

All the in vitro experiments were independently repeated at least three times. Data are presented as the mean ± standard error of the mean. For the in vivo experiments, *t*-tests and Mann–Whitney U-tests were used to analyze the differences in biodistribution between the NP and nanoconjugate groups for each tested organ. Differences between groups were considered significant at *p* < 0.05. All statistical analyses were performed using Prism 8 (GraphPad Software Inc., La Jolla, CA, USA).

## 3. Results

### 3.1. Physicochemical Characterization and In Vitro Release Profile of Lenva-NPs and Lenva-NPs-Cetuximab

The lenva-NPs examined here were characterized in terms of size, PDI, zeta potential, and morphology (Figure 1). It has been shown that the incorporation of lenvatinib into PLGA NPs resulted in an increase in particle size from 222.9 to 262.4 nm (Appendix A, bars). Moreover, the addition of lenvatinib resulted in a slight increase in dispersity value (Appendix A) from 0.12 to 0.19. In comparison with the zeta potential of the blank PLGA nanoparticles (−16 ± 2.1 mV), a small decrease in the mean zeta potential of the lenva-NPs (−20.9 ± 2.9 mV) was seen after the addition of lenvatinib to the PLGA NPs (Appendix A, dots). Since the development of the lenva-NPs was successful in terms of particle size, PDI, zeta potential, morphology, and encapsulation efficiency, we chose to include the monoclonal antibody cetuximab on the surface of the NPs to target cells that overexpress the EGFR. To obtain lenva-NPs-cetux, a method called covalent binding, which was based on EDC/sulfo-NHS cross-linking chemistry, was utilized to adsorb monoclonal antibody cetuximab on the surface of the lenva-NPs. This reaction couples the primary amino groups with carboxylates to form stable amide crosslinks. Surface plasmon resonance, flow cytometry, and ELISA can all be used to confirm the presence of cetuximab on lenva-NPs, although each has its own advantages. Unbound cetuximab in the supernatant was detected via spectrophotometry using a colorimetric microBCA protein assay kit. According to the findings of our study, 76% of the cetuximab added was distributed across the surface, which is equivalent to 18.1 μg of cetuximab per mg of lenva-NPs (Figure 2). The dispersion value of the lenva-NPs-cetux (Figure 1B) was very similar to that obtained for the lenva-NPs. This suggests that the addition of cetuximab did not result in a modification of the NP structure. When observed under TEM, the lenva-NPs and lenva-NPs-cetux (Figure 1C,D) displayed a spherical shape and a uniformly smooth surface, indicating that the addition of cetuximab did not result in a change in the morphology of the NPs. Furthermore, the TEM observations agreed with those obtained using dynamic light scattering. Thus, an NP’s in vitro and in vivo performance can be influenced by its surface shape, which should be smooth and free of holes. This also indicates that the solvent was successfully evaporated. The encapsulation efficiency of lenvatinib (79 ± 6.4%) was determined with the RP-HPLC method using supernatant obtained during the preparation process. Under our experimental conditions, the components produced after dissolving the NPs in organic solvents limited the detection of the lenvatinib content by HPLC due to interfering peaks. Lenvatinib encapsulated in PLGA NPs is mainly attributed to the partition coefficient and is therefore retained in the organic phase when the microspheres solidify; however, the encapsulation also depends on many other aspects, such as the NPs’ size, the lipophilicity of the drug incorporated, and the preparation method. The in vitro lenvatinib released from the NPs in vitro equated to 9% after 24 h (Figure 3).

### 3.2. Physicochemical Characterization of DilC 18-NPs and DilC 18-NPs-Cetuximab

To determine whether the incorporation of the DilC 18 fluorescent dye would affect the physicochemical parameters of the NPs that control cellular uptake, we determined the particle size, zeta potential, and morphology of the DilC 18 loaded PLGA NPs (DilC 18-NPs) and cetuximab-conjugated DilC 18 PLGA NPs (DilC 18-NPs-cetux) (Figure 4). When compared with the lenva-NPs, a slight increase in particle size from 262 nm (Figure 1A) to 271.7 nm (Figure 4A), as well as similar values in zeta potential of −20.9 mV for lenva-NPs (Figure 1A) and −19.8 mV for DilC 18-NPs (Figure 4A), were observed. Regarding particles with cetuximab adsorbed on the surface, the values for size and zeta potential were similar after substituting lenvatinib with DilC 18: lenva-NPs-cetux (size 290.6 nm, zeta −16.3 mV, Figure 1A) compared with DilC 18-NPs-cetux (size 291 nm and zeta −15.4 mV, Figure 4A).

### 3.3. In Vitro Cytotoxicity Studies of Lenva-NPs and Lenva-NPs-Cetuximab

To evaluate the effectiveness of both nanoparticle formulations in vitro before considering their application in the in vivo experiments, we performed an MTT assay using two thyroid cell models, CAL-62 and Nthy-ore 3-1, which exhibit different levels of expression of EGFR (high and low, respectively) [25]. It is important to note that there is a lack of data available on the cytotoxic impact of the drug when encapsulated in nanoparticles. For that reason, both cell lines were exposed to the required amount of nanoparticles loaded with 20 μM lenvatinib for 24 h; this dose/time frame is within the effective range reported for anaplastic thyroid cancer cell lines [29,30]. The results after treatment with both NPs demonstrated a cytotoxic effect in both cell lines, despite slight variations in cell viability depending on the cell line and drug formulation administered. In CAL-62, the effect of the lenva-NPs-cetux was higher (46.8 ± 2.5%), reaching IC50 in comparison with the lenva-NPs (54.5 ± 1.3%). In the Nthy-ori-3-1 cell line, both NPs produced cytotoxicity in the cells; however, cytotoxicity was marginally decreased in comparison with the CAL-62, and neither formulation reached IC50. (lenva-NPs-cetux 55 ± 6.7% versus lenva-NPs 61 ± 1.6%). Although no significant differences between the NPs were detected in either cell line, there is a tendency to produce a higher effect when the NPs are able to target the EGFR (Figure 5A,B).

### 3.4. Biodistribution of Lenva-NPs and Lenva-NPs-Cetuximab

The intragastric administration showed high radiant efficiency signal uptake in the tumors at 5 h with the targeting NP formulation, without significant accumulation in the rest of the organs analyzed, except in the kidney and lung at 5 h with lenva-NP-cetux and at 48 h with lenva-NP (Figure 6).

### 3.5. In Vivo Evaluation of the Therapeutic Effect of Lenva-NPs and Lenva-NPs-Cetux

The in vivo treatment via intragastric administration, three times weekly for four weeks, did not produce negative effects in the weight of the mice in comparison with the animal controls; no significant differences were observed between both formulations. (Figure 7A). Moreover, normal activity without lethargy or apathy after intragastric NP administration was observed. The effect on tumoral growth was evaluated by measuring tumoral volume (mm^3^) four weeks after beginning the treatments (lenva-NPs and lenva-NPs-cetux). The results showed that both formulations were able to cause a significant reduction in tumoral volume by approximately 30% (*p* = 0.05) compared with controls, without observed significant differences between the NPs (Figure 7B). The ex vivo analysis via macroscopic observation of the tumoral tissues detected that necrosis was higher in the tumors of animals treated with both formulations (20.58 ± 6.43% and 22.91 ± 12.60% with and without cetux, respectively) in comparison with controls (4.91 ± 5.55%), without significant differences between the formulations (Figure 7C). As shown in Figure 7D, histological analysis of the non-tumor-bearing organs displayed no significant histological changes in key tissues, such as the kidney, liver, spleen, lung, pancreas, and brain of the mice treated with both NPs, in comparison with controls. Moreover, no toxic effects were observed in the animals, since they maintained normal activity without lethargy or apathy after intragastric NP administration throughout the entire experiment. The histological analysis of the necrosis in the tumors (Figure 8A) confirmed the macroscopic results described previously. The percentage of necrosis (stromal fibrosis) found in the tumoral tissue was sharply higher in those animals treated with the NPs loaded with the drug (72.5 ± 7.3% lenva-NPs; 67 ± 10.7% lenva-NPs-cetux) than in the control animals (22.9 ± 9.89%; *p* = 0.001; Figure 8B). Nevertheless, no significant increments in the necrotic features were found when NPs decorated with cetuximab were used. In contrast, the evaluation of the percentages of apoptotic/karyolytic forms showed very significant differences in the percentage of cells with reactive changes in the cytoplasm or nucleus, which are compatible with cell death/necrosis, between control tissues (6.3 ± 4.1%) and tissue samples treated with NPs (*p* = 0.0001). Moreover, the percentage of morphological changes was higher in the NPs decorated with cetuximab versus non-decorated NPs (85 ± 19.26% and 44.2 ± 9.57%, respectively; *p* = 0.003; Figure 9A,B). In addition, the mitotic index count/field used to determine proliferative cellular arrest showed that this parameter was lower in both NPs (11.4 ± 5.68% lenva-NPs; 5.71 ± 3.25% lenva-NPs-cetux; *p* = 0.001) compared with the controls (45.4 ± 14.64%) (Figure 9A,B). In order to validate the proliferation rate within tumors of animals subjected to NP treatments compared with control groups, the assessment of Ki-67 expression demonstrated a decreased number of positive cells in tumors from animals treated with both NPs-lenva (39.7 ± 13.44%) and NPs-lenva-cetux (39.143 ± 6.70%) when compared with control (84.16 ± 9.31%). However, no significant differences were observed between decorated and non-decorated NPs (Figure 10A). Interestingly, cytochrome C expression was higher in tumors treated with lenva-NPs compared with decorated NPs. However, the expression of AIF was higher in tumors treated with both NPs (lenva-NPs and Lenva-NP-cetux) compared with control (Figure 10B,C). The analysis of the neovascularization (i.e., vascular density and vascular length density) in the tumors of the animals treated with NPs with and without cetuximab targeting showed a significant reduction in these two parameters in comparison with the controls, although there were no significant differences between them (Figure 11A–D).

## 4. Discussion

This study utilized a PLGA vector system for lenvatinib administration, evaluating whether anti-EGFR targeting could improve the efficiency of delivery and mechanistic action. However, the efficacy of drug delivery systems is heavily dependent on the physicochemical features of the resulting NPs prior to surface decoration with the desired target ligands. For example, the particle size and PDI have a direct impact on the physical stability, cellular uptake pathway, drug release, and biodistribution of NPs; hence, these parameters deserve special consideration when formulating NPs. Moreover, a variety of variables, such as the surfactant and polymer composition, homogenization method, and solvent viscosity, also influence particle size and PDI. As a result, the adaptability of the system allowed us to manage the final size and PDI under the specified requirements. The incorporation of lenvatinib into the NPs appeared to affect the free carboxylic end groups on the NP surface. Curiously, large zeta potential values (either positive or negative), typically more than 20 mV, prevent aggregation by electrostatic repulsions in nanosuspension. On the other hand, we observed that stabilized suspensions at zeta potentials of about 10 mV are possible. It is necessary to avoid aggregations that may be quickly cleared by macrophages before the drug reaches the target cells to prolong the circulation period of NPs in the bloodstream. It is generally recognized that negatively charged NPs do not interact with blood proteins. Furthermore, electrostatic interactions with negatively charged cellular membranes make it easier for positively charged systems to enter cells. For this reason, the optimum value of zeta potential depends greatly on the final application. In this sense, our data match the reports of other authors’ reports [31,32]. Interestingly, when cetuximab was incorporated, as shown in the bars in Figure 1A, we detected an increase in particle size from 262 nm (lenva-NPs) to 289 nm (lenva-NPs-cetux) and a decrease in zeta potential from −20.9 mV to −16.3 mV (Figure 1A, dots). The increase in particle size after cetuximab binding could be explained by the antibody’s demand for more space in the formulation, as well as the neutralization of the carboxylate groups in the PLGA polymer, which results in less negatively charged systems. With respect to the in vitro release profile of lenva-NPs, the effect detected could be attributed to the diffusion of the medication that had been adsorbed or weakly attached to the NPs. In contrast, the amount of drug incorporated into the NPs determines the rate of progressive release—which was noted over 20 days in this study. The low solubility of lenvatinib could be responsible for the slow release rate observed. An increased release of lenvatinib was noticed over the next 10 days, which was likely caused by the degradation of the PLGA matrix throughout the later phases. Consequently, the degradation rate and release profile were found to be affected by the content of the polymer matrix. By the end of the experiment, which lasted 38 days, the NPs had released about 15% of the loaded lenvatinib—probably due to the limited solubility of lenvatinib on the release medium. These results are comparable with the data showing the efficacy of lenvatinib-eluting microspheres in preclinical studies of hepatocellular carcinoma [33].

Moreover, the cumulative release sustained over time detected in lenva-NPs could be important for clinical use because it could partially avoid the side effects associated with high concentrations of the drug. In epithelial tumoral cells, EGFR is frequently mutated and/or overexpressed. This event has been used as a therapeutic strategy using specific antibodies that block the ligands by binding to the extracellular domain of this receptor, and, as a consequence, induce a cellular signaling cascade arrest. Moreover, the tumoral targeting and therapeutic efficiency of anti-EGFR-NPs for delivering anticancer drugs have been investigated in vitro and in preclinical studies, with good results [34,35,36,37]. The results suggest that for tumors that overexpress specific receptors, such as EGFR as demonstrated in glioblastoma, the antibody-conjugate in NPs can be advantageous in drug delivery and can improve the antiproliferative efficacy in the tumor cells [34].The lack of significant differences in the in vitro cytotoxic effect of both formulations, as shown in Figure 5, can be attributed to the relatively short duration of the assay, which was limited to 24 h. This observation is consistent with the results of in vitro drug release experiments, which indicate that lenvatinib is gradually released over a 10-day period due to the progressive degradation of the PLGA matrix over time. It is worth noting that there are currently no available data regarding the encapsulation of this drug within PGLA nanoparticles. Nevertheless, the biodistribution analysis of the nanoparticles revealed that the Lenva-cetux NPs were more efficiently taken up by tumors 5 h after administration compared with the Lenva-NPs. We must emphasize that, despite the lack of knowledge regarding the absorption efficiency via intragastric administration of PLGA nanoparticles used in the in vivo experiments, the results are significant. This is because the nanoparticles were able to maintain stability within the stomach and exhibited the capability to cross the intestinal barrier, potentially achieving their therapeutic goals [38,39]. The histological analysis (apoptosis/karyolysis) and the evaluation of mitosis in the tissue samples indicate that the use of cetuximab-decorated nanoparticles could offer advantages compared with non-decorated nanoparticles. Nevertheless, some reports demonstrate that the use of full-size antibodies as a ligand does not improve their therapeutic effect, owing to their high immunogenicity or the intrinsic tumoral resistance to EGFR inhibitors, proposing the decoration of NPs with an EGFR-aptamer [40,41]. Lenvatinib affects the inhibition of VEGFRs, promoting their anti-tumor activity by reducing the density of microvessels and increasing tumor necrosis [42]. The analysis of neovascularization (vascular density and vascular length density) in the tumors of the animals treated with NPs with and without cetuximab-targeting showed a significant reduction in these two parameters in comparison with the controls. These results indicate a high degree of ischemia in the tumors from animals treated with both NP formulations, although there were no significant differences between them. Moreover, lenvatinib is an inhibitor of multiple receptor tyrosine kinases capable of inducing early processes of apoptosis and necrosis in Huh-7 cells [43]. Furthermore, it has been described that PLGA NPs loaded with temozolomide conjugated with cetuximab, designed to target cancers that overexpress EGFR, were able to promote late processes of apoptosis and necrosis compared with PLGA NPs loaded with temozolomide [44] Although our Ki67 data did not reveal significant differences between both formulations in the tissue areas far from the necrotic areas of the tumors, we must highlight that in the analysis of cells with typical morphologies of nuclear apoptosis (condensation and fragmentation), as well as the morphology of karyolysis, we demonstrated that the tumoral tissues obtained from animals treated with decorated nanoparticles showed a greater percentage of these cell types compared with those treated with undecorated nanoparticles. The key initial step of the apoptosis process is closely associated with mitochondrial dysfunction and the rapid release of two proapoptotic proteins, alongside the release of cytochrome c (cyt c) and apoptosis-inducing factor (AIF). Although these two proteins operate differently in the apoptotic pathway (caspase-dependent and caspase-independent, respectively), their release from the mitochondria triggers chromatin condensation, DNA fragmentation, and death in cells [45,46]. In this context, our data regarding the morphology of apoptotic cells and karyolysis, in relation to the results of cytochrome c and AIF protein expression, indicate the capability to activate both mechanisms in tumors treated with both formulations. However, the distinct pattern observed between them may suggest that decorated NPs have a more rapid ability to induce a greater level of cell injury, with a loss of proapoptotic protein expression, even though a percentage of the tumoral tissue still remains viable.

In summary, this proof-of-concept study demonstrated that both formulations can maintain their therapeutic effect after intragastric administration, indicating stability in the digestive system and the ability to cross the intestinal barrier. However, cetuximab-decorated NPs show potential advantages in uptake capacity *in vivo*, and greater ability to initiate apoptosis /necrosis, suggesting their greater effectiveness. Further investigations will be necessary to fully understand the specific apoptosis mechanisms they promote.

Finally, the good tolerance observed in animals with these treatments suggests that these delivery systems may offer advantages over conventional methods and open the door to exploring other TKI drugs.

## Figures and Tables

**Figure 1 biomolecules-13-01647-f001:**
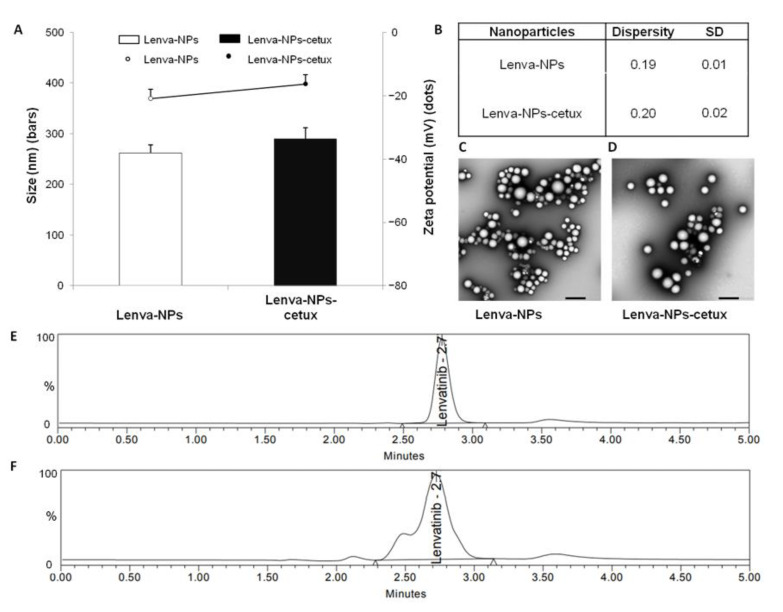
Physicochemical characterization of lenva-NPs and lenva-NPs-cetux. (**A**) Size (bars) and zeta potential (dots); (**B**) Dispersity index and standard deviation values of lenva-NPs and lenva-NPs-cetux. Each value represents the mean ± standard deviation of three independent measurements; (**C**) TEM images of lenva-NPs. Scale bars: 500 nm; (**D**) TEM images of lenva-NPs-cetux. Scale bars: 500 nm. Representative RP-HPLC chromatograms for lenvatinib: (**F**) a calibrator sample spiked with 0.5 µg/mL lenvatinib; (**E**) a PLGA-lenva-NP sample 4 days after the start of the release study.

**Figure 2 biomolecules-13-01647-f002:**
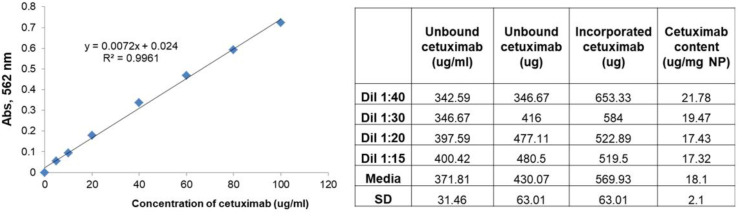
Determination of cetuximab content with colorimetric micro BCA assay.

**Figure 3 biomolecules-13-01647-f003:**
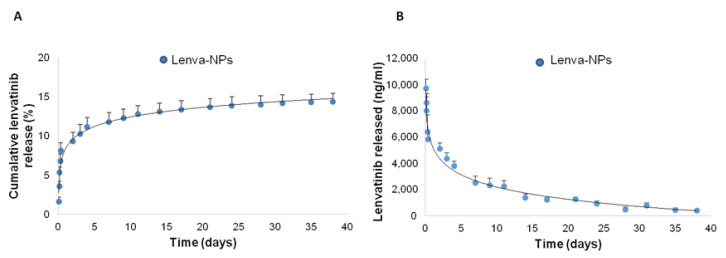
In vitro lenvatinib release from lenva-NPs. (**A**) Time course of the cumulative percentage of lenvatinib released, (**B**) lenvatinib released over time. Values are represented as the mean (± standard deviation, SD) from three batches of each formulation.

**Figure 4 biomolecules-13-01647-f004:**
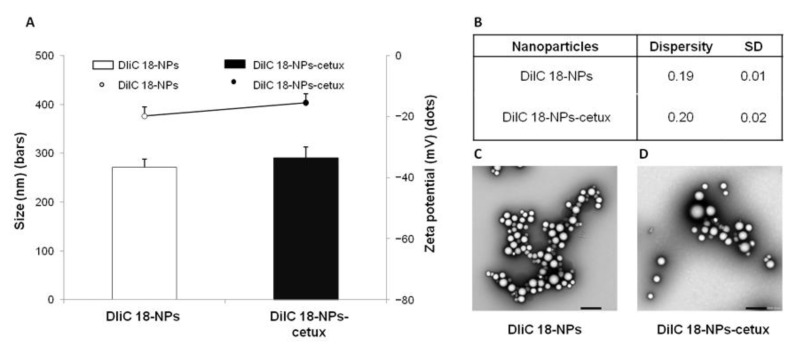
Physicochemical characterization of DilC 18-lenva-NPs and DilC 18-Lenva-NPs-cetux. (**A**) Size (bars), and zeta potential (dots). (**B**) Dispersity index and standard deviation values of DilC 18-lenva-NPs and DilC 18-lenva-NPs-aEGFR. Each value represents the mean ± standard deviation of three measurements. (**C**) TEM images of DilC 18-Lenva-NPs. Scale bars: 500 nm. (**D**) TEM images of DilC 18-Lenva-NPs-aEGFR. Scale bars: 500 nm.

**Figure 5 biomolecules-13-01647-f005:**
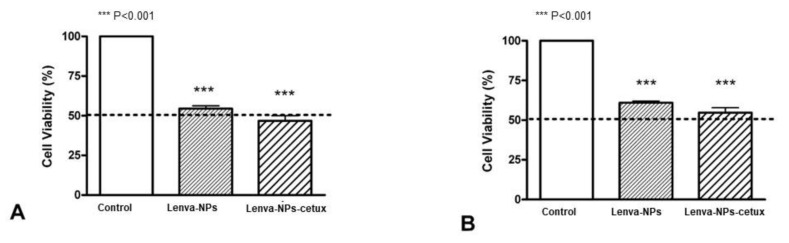
In vitro cytotoxicity assays of lenva-NPs and lenva-NPs-cetux in two thyroid cellular models. (**A**) CAL-62 and (**B**) Nthy-ori3-1. The percentages of cell viability were normalized to the control cell cultures in completely particle-free media. Each data point represents mean ± standard deviation (N = 3). The statistical significance was calculated using two-way ANOVA followed by a Bonferroni Multiple Comparisons test (*** *p* < 0.001).

**Figure 6 biomolecules-13-01647-f006:**
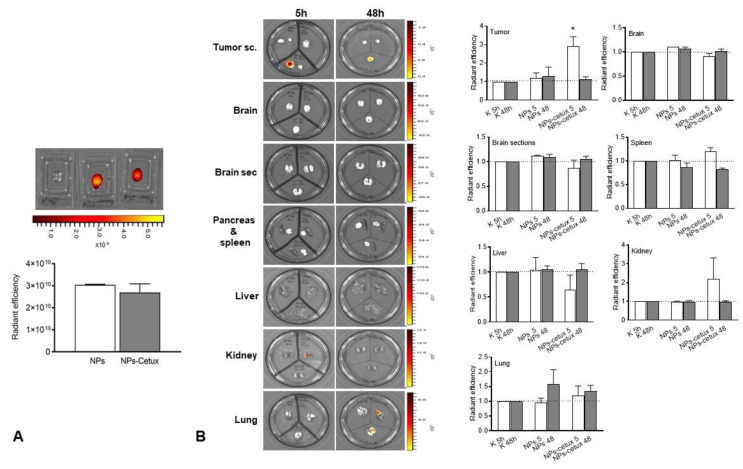
Biodistribution profiles of DiIC-loaded nanoparticles in tumor-bearing mice. DiIC NPs were administered via intragastric administration. (**A**) Histogram of the radiant efficiency of the NPs and NPs-cetux. (**B**) Ex vivo fluorescence imaging of major organs at 5 h and 48 h after injection of NPs and NPs-cetux in comparison with vehicle (K) and representative histogram of their radiant efficiency. Two-way ANOVA followed by a Bonferroni Multiple Comparisons test (*p* * < 0.05).

**Figure 7 biomolecules-13-01647-f007:**
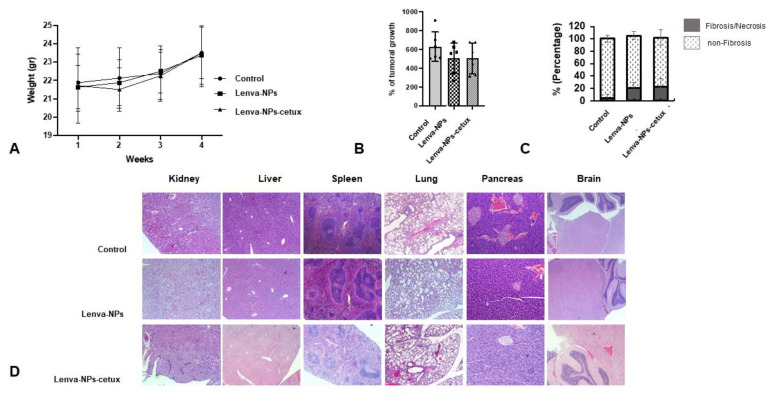
Assessment of in vivo tumor growth and response to lenva-NPs and lenva-NPs-cetux. (**A**) Timeline and body weight results of the experimental procedure. (**B**) Percentage of tumoral growth at the end of the experiment. (**C**) Histograms showing the percentage of fibrosis/necrosis and non-fibrosis score in the tumoral tissue at the end of the experiment. (**D**) Histological section stained with hematoxylin and eosin (H&E) for histopathological analysis of the main organs of the xenograft tumor-bearing mice after intragastric treatment with PBS and NPs encapsulated with lenvatinib, with or without cetuximab decoration (NPs). H&E; magnification 40×. Data represent means ± SD (N = 5).

**Figure 8 biomolecules-13-01647-f008:**
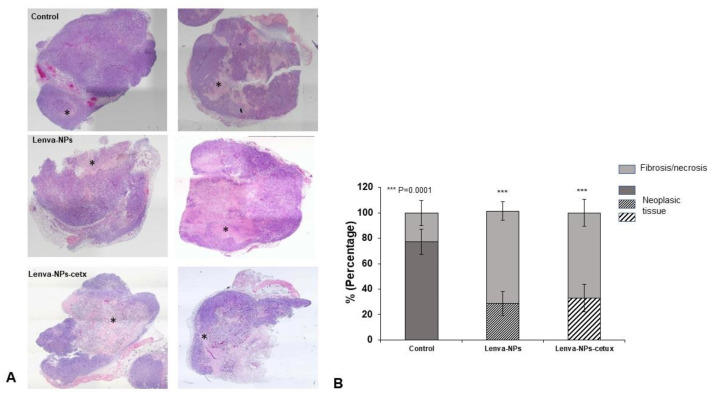
Tumor necrosis after NP treatment. (**A**) Representative H&E-stained tumor sections from the control; lenva-NPs and lenva-NPs-cetux groups. (**B**) Percentage of necrosis with H&E staining (one-way ANOVA followed by Holm–Sidak post hoc analysis). The statistical significance was calculated using two-way ANOVA followed by a Bonferroni Multiple Comparisons test; *** *p* = 0.0001, between treated animals versus control. * Corresponds to necrotic areas. Magnification 4×.

**Figure 9 biomolecules-13-01647-f009:**
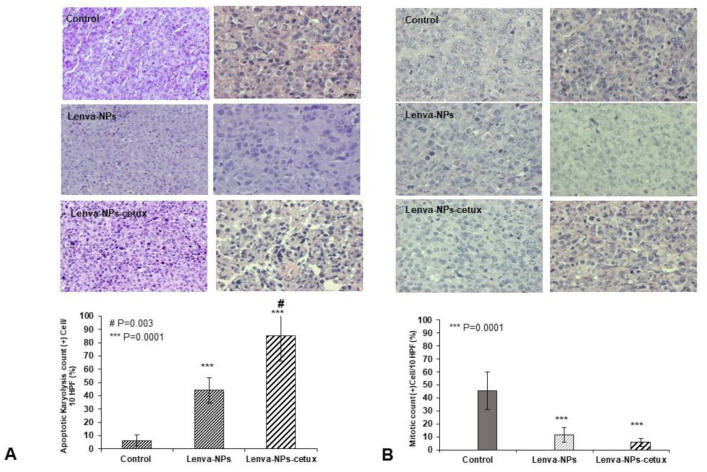
Histology analysis with H&E staining of xenografted tumors. (**A**) Representative images (superior panel) and quantification of results displaying the % of apoptotic/karyolitic bodies in the animals treated with both NPs in comparison with control (inferior panel). (**B**) Representative images (superior panel) and quantification of results displaying the % of mitotic index in the animals treated with both NPs in comparison with control. The statistical significance was calculated using two-way ANOVA followed by a Bonferroni Multiple Comparisons test. Magnification 40×.

**Figure 10 biomolecules-13-01647-f010:**
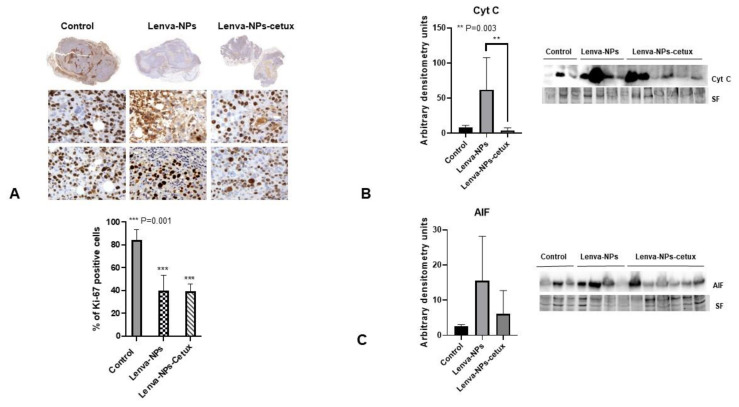
Immunohistochemical evaluation of proliferation and Western blot of apoptosis markers in tumors. (**A**) Immunostained (Anti-Ki-67 antibody) tumor sections from control; lenva-NP and lenva-NP-cetux groups and quantification of the percentage of positive cells in 5 randomly selected regions in each tumor section under 400× magnification in 4 animals in each group. Data are expressed as mean ± SD. *** *p* < 0.001 compared with control. The Ki-67 positive and total numbers of cells were counted in 5 randomly selected regions in each tumor section under 400× magnification, and the percentage of positive cells was calculated. (**B**,**C**) Representative Western blot and quantification of cytochrome c and AIF of total protein from tumors of control, lenva-NP and lenva-NP-cetux groups, histograms correspond to the densitometric analysis.** *p* = 0.003. Original western blots can be found in Appendix A.

**Figure 11 biomolecules-13-01647-f011:**
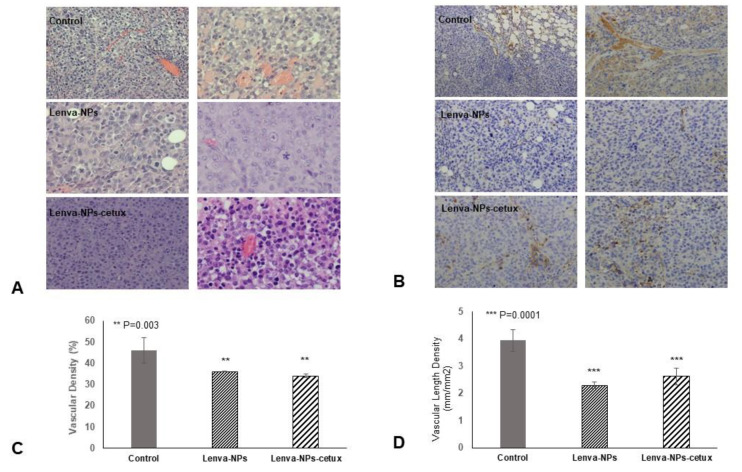
Histology and immunohistochemical evaluation of vascular architecture in tumors. (**A**,**B**) Representative H&E and immunostained (Anti-VEGF Receptor 2 antibody) tumor sections from control, lenva-NP, and lenva-NP-cetux groups; (**C**,**D**) Quantification of the percentage of vascular density was processed using Vessel Analysis plugin for Fiji software 2.9.0 (http://imagej.net/Vessel_Analysis, accessed on 1 August 2023) (**C**) and the vascular length density number of vessels (**D**). Results are shown as mean ± SD of three sections of six mice per group. The statistical significance was calculated using two-way ANOVA followed by a Bonferroni Multiple Comparisons test. Magnification 40×.

## Data Availability

Data is contained within the article or Appendix A.

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
