# Peer review of "Lenvatinib-Loaded Poly(lactic-co-glycolic acid) Nanoparticles with Epidermal Growth Factor Receptor Antibody Conjugation as a Preclinical Approach to Therapeutically Improve Thyroid Cancer with Aggressive Behavior"

_biomolecules, 2023, doi:10.3390/biom13111647_

Round 1
Reviewer 1 Report
Comments and Suggestions for Authors
The tyrosine kinase inhibitor lenvatinib has been approved for the treatment of aggressive radioactive iodine-refractory differentiated thyroid cancer, but it is associated with various adverse effects. The use of nanoparticles could allow for targeted drug delivery, reducing some side effects. This study assessed in vitro and in vivo the cytotoxicity, biodistribution, and therapeutic efficacy of lenvatinib-loaded PLGA nanoparticles (NPs), both with and without decoration using antibody conjugation (cetuximab), as a novel therapeutic approach for managing aggressive thyroid tumors, through two cellular models, normal thyroid cells (Nthy-ori 3-1) and anaplastic thyroid cells (CAL-62). The NPs demonstrated success in terms of particle size, polydispersity index (PDI), zeta potential, morphology, encapsulation efficiency, and cetuximab distribution across the surface. In vitro analysis revealed cytotoxicity in both cellular models with both formulations, but only the decorated NPs achieved an ID50 value in CAL-62 cells. Biodistribution analysis following intragastric administration in xenografted thyroid mice demonstrated good stability in terms of intestinal barrier function and tumor accumulation. Both formulations were generally well-tolerated and increased tumor necrosis; however, decorated NPs exhibited enhanced parameters related to apoptotic/karyolytic forms, mitotic index, and vascularization compared to NPs without decoration. The Authors conclude that these findings suggest a promising strategy for administering TKIs in a more targeted and effective manner.
The paper is interesting, I suggest only some minor points:
1. On line 85, what does “DTS” mean? Was it “DTC”?
2. English language and typing errors should be revised, see for example: “… in vitro their cytotoxicity and; as …” on line 88
3. I suggest to improve the Introduction section, regarding the main molecular targets and tyrosine kinase inhibitors in aggressive dedifferentiated thyroid cancer, also citing recently published papers, such as 10.21037/gs.2019.10.18 and 10.3389/fonc.2022.1099280
4. In the study only two thyroid cell lines were evaluated, while it could be interesting to assess also primary cell cultures, since their characteristics are quite different from those of continuous cells. Discuss this point.
Comments on the Quality of English Language
English language and typing errors should be revised, see for example: “… in vitro their cytotoxicity and; as …” on line 88
Reviewer 2 Report
Comments and Suggestions for Authors
The manuscript by Revilla et al. describes the study of lenvatinib-loaded PLGA NPs conjugated with EGFR antibody as preclinical approach for thyroid cancer treatment.
Here some points that I would like to be addressed.
- please rephrase lines 82-84
- explain what cetuximab is as you introduce it first time in the main text; explain a bit more its function and the previous studies
- line 88 rephrase
- in section "preparation of dilc 18 fluorescently labelled nanoparticles" authors cited section 2.2, 2.5 and 2.5.1, but there is no division in sub-section in this part, so please check it
- section "in vitro cytotoxicity studies of lenva-nps and lenva-nps-cetux". It is described that MTT was directly added at the end of the incubation. Do this mean that no washing has been done to remove not internalized/interacted NPs from medium? Have you also checked for the interaction between NPs and MTT in terms of absorbance? It is well known that NPs can absorb and interfere with many assays like MTT assay. A control comprising NPs alone should be added to confirm that they are not producing any interference with the assay signal.
- fig 1 B, tem images, please make the scale bar more visible
- why you put supplementary figures in the main text?
- I suggest you to use TEM images with the same magnitude in fig. 1 and in fig. 4. Preferentially the magnitude of fig. 4 for both.
- section 3.3. Why did you select the 20 uM as concentration of NPs? This is a very key point, so it should be explained very well. Have you tried different concentrations? Did you choice this concentration based on previous published results? What about other concentrations?
- section 3.4 The authors introduce here the results of NP intravenous administration, but they didn't talk about this administration before in the text, neither in methods section.
Also because of that, in my opinion section "biodistribution of np and nps-cetus nanoparticles" should be rearranged to make it more clear for the reader.
Also explain better why you selected the intragastric administration.
- section 3.4 the authors said the after intravenous administration they notice an unexpected high accumulation in important organs. Really? It should be obvious that after this kind of injection, do to the blood circulation and distribution, the NPs will accumulate in such organs. In fact, the authors referred to this in the discussion, line 662 and so on.
- fig.7 B there is no timeline of experiment in this figure
- In my opinion this work shows that NPs decorated with cetuximab work in a very similar manner compared to the not-decorated onces.
I think that the authors made a very confused discussion that don't address the key point: why it should be more useful to employ these decorated NPs since the results don't show a clear advantage? There is a very low difference in the effect on in vitro cytotoxicity (fig. 5) between the two NPs and between the two different cell line that have a different receptor expression. The efficiency of a NPs decorated with an antibody to target the respective receptor should be much more that what is for these NPs. Why there is this low efficiency in targeting?
For most of the experiments, the two NPs, decorated or not, shown the same behavior, a part from apoptosis (fig9) and cyt c (fig10). Also in this case, the authors didn't explain the possible reason for this effect.
In my opinion there is no clear effect of decorated NPs that justify the statement of the authors.
Round 2
Reviewer 2 Report
Comments and Suggestions for Authors
The authors addressed all my questions and followed my suggestions. In my opinion the paper can be accepted in the current form